# 16 Weeks of Physically Active Mathematics and English Language Lessons Improves Cognitive Function and Gross Motor Skills in Children Aged 8–9 Years

**DOI:** 10.3390/ijerph192416751

**Published:** 2022-12-13

**Authors:** Ruth Boat, Simon B. Cooper, Fabio Carlevaro, Francesca Magno, Giulia Bardaglio, Giovanni Musella, Daniele Magistro

**Affiliations:** 1Sport, Health, and Performance Enhancement Research Centre, Department of Sport Science, School of Science and Technology, Nottingham Trent University, Clifton Campus, Clifton Lane, Nottingham NG11 8NS, UK; 2Polo Universitario Asti Studi Superiori (Uni-Astiss), 14100 Asti, Italy; 3University of Torino, Torino, Italy

**Keywords:** physically active lessons, cognitive function, gross motor skills, physical activity, learning

## Abstract

The aim of the present study was to examine the effects of physically active lessons, implemented through the Mathematics and English Language curriculum, on cognitive function and gross motor skill development. Following ethical approval, 192 children aged 8–9 years were randomly allocated to an intervention group (*n* = 98) or a control group (*n* = 94). The intervention consisted of 8 h.wk^−1^ of physically active lessons, equally split between Mathematics and English Language, for 16 weeks. Cognitive function (digit span, coding and arithmetic reasoning) and gross motor skill development (TGMD-3) were assessed at baseline and follow-up. The improvement in every domain of cognitive function was greater in the intervention group compared to the control group (group * time, *p* = 0.008–0.023, *d* = 0.34–0.42). Furthermore, total TGMD-3 score (group * time, *p* < 0.001, *d* = 1.16) and both sub-scales (locomotor, *p* < 0.001, *d* = 0.63; object control, *p* < 0.001, *d* = 1.29) also improved by a greater extent in the intervention group than in the control group. These findings suggest that 16 weeks of physically active lessons, taught in both Mathematics and English Language curriculum, synergistically improved cognitive function and gross motor skill development in primary school children aged 8–9 years.

## 1. Introduction

Despite the well-documented benefits of physical activity for physical, mental, and social well-being in young people being widely reported and accepted [1,2], it is also well-documented that many young people do not meet the recommended 60 min of moderate to vigorous physical activity per day [3]. For example, it has been reported that more than 50% of young people worldwide fail to reach the recommended 60 min moderate to vigorous activity per day [4], with more recent data from Italy suggesting that as few as 7% of girls and 18% of boys aged 11 years achieve the recommended physical activity targets [5]. To address this issue, it has been proposed that schools are a key intervention target for increasing physical activity in young people [6,7]. However, due to the focus of the education sector primarily on educational outcomes (rather than health and well-being outcomes), opportunities (and time allocation) for physical activity are often reduced [8,9]. One potential way to maximize time for academic subjects alongside physical activity is their combination within the classroom [10], typically referred to as physically active lessons.

Physically active lessons are teacher-implemented academic lessons that incorporate moderate–vigorous student movement, typically used to teach new knowledge that is related to already well-known concepts [11]. Recent systematic reviews and meta-analyses have suggested that the inclusion of physically active academic lessons is an effective intervention strategy to increase physical activity [12], enhance academic achievement/educational outcomes [12,13], and improve classroom behavior [14].

However, the evidence base currently lacks consistency, potentially due to the variability in intervention design and the outcomes observed [15]. For instance, although enhanced cognitive function has been suggested as a potential underlying mechanism explaining the beneficial effect on academic achievement, the effects of physically active lessons on cognitive function remain unclear. Reasons for this could be attributed to the fact that the measurement of cognitive function can be challenging due to the various domains of cognitive performance assessed, as well as the diversity of methods used to measure the construct [12,13,15]. Encouragingly, recently, it was reported that two years of physically active lessons improved multiple domains of cognitive performance in 6–7 year-old children (attention, memory, and verbal fluency) when compared to a control group [16], suggesting that enhancements to academic achievement may well be, at least in part, mediated by improvements in cognitive performance. Future work should, however, consider the impact of physically active lessons on cognitive performance across the age range of primary school children, particularly given that age is a potential moderating factor in the physical activity–cognition relationship [17].

Moreover, the incorporation of physical activity bouts within the classroom has been suggested to enhance motor skill development [16]. Gross motor skills are commonly defined as goal-directed movement patterns including large whole-body movements, whole body stretches, and locomotion [18,19], and are imperative for psychosocial development, health, and well-being [20,21]. Research has shown that gross motor skill development is enhanced in young people who engage in increased physical activity [22]. However, to date, limited research has explored the effect of the incorporation of physical activity within the classroom on motor skill development in young people. Recently, Magistro et al. [16] found that physically active Mathematics lessons led to greater improvements in overall gross-motor skill development over a two-year period, in the intervention group, compared to the control group, in primary school children aged 6–7 years old. This suggests that physically active Mathematics lessons have the potential to accelerate gross motor skill development. However, it is worth noting that within this study, physical activity was only incorporated into Mathematics lessons. It is currently unknown whether the inclusion of more physically active lessons (e.g., Mathematics and English Language) within the school curriculum could lead to greater influences on motor skill development.

Therefore, the aim of the present study was to explore how the incorporation of physical activity bouts within the school curriculum (in Mathematics and English Language lessons) influences cognitive function and motor skill development across sixteen weeks in primary-school children. Considering previous literature (e.g., [12,13,15,16]), it was hypothesized that incorporating physical activity simultaneously with Mathematics and English Language lessons would lead to enhanced motor skills and cognitive function, compared to a control condition. The present study adds to the evidence base by evaluating the effects of physically active lessons within both Mathematics and Language lessons, in older primary school children than those studied previously (e.g., [16]) and with both cognition and gross motor skills as outcome variables.

## 2. Methods

### 2.1. Design

Prior to the commencement of data collection, the study was approved by the ethical committee of the University of Torino (ID 100949). Assent was provided by the children themselves and informed consent was obtained from children’s parents/guardians. In a between-subject experimental research design, 6 classes were allocated to the intervention group and 6 matched classes were allocated to the control group. The intervention consisted of integrating physical activity bouts in all Language (second language English) (4 h per week) and Mathematics (4 h per week) lessons, implemented for 16 weeks (see Figure 1). The intervention was co-designed with the English Language, Mathematics, and Physical Education teachers, and in line with the National Curriculum Guidelines for Primary Schools in Italy. All activities were implemented within the classroom by the usual class teachers. The control condition involved the typical Mathematics and English Language National Curriculum.

The 16-week intervention involved a number of games within the physically active lessons, each linked to a specific Mathematical or English Language component of the curriculum. Each lesson contained a warm-up, an explanation, two main activities, and a summary. Each lesson started with a warm-up, which comprised walking, jogging running, walking and/or other Physical Education/games-based activities (~10 min). Following the warm-up, the teacher explained the Mathematics/English Language concept being taught (~15 min). Subsequently, the physically active learning task was explained and demonstrated by the teacher (~15 min). Children then participated in the physically active learning task (~10 min). For variety, there was then alternative version of the physically active task (~10 min). Finally, in the review, the teacher summarized the learning that had taken place during the lesson (~5 min).

Throughout the 16-week intervention, the focus was on the reinforcement of a number of key Mathematical and English Language concepts that were previously taught, as this is when physically active lessons are suggested to be most effective [11]. For example, in Mathematics, topics included understanding, reading, and writing numbers in decimal notation, as well as performing simple calculations, including addition and subtraction of numbers <10. Furthermore, in English Language, the topics included listening and comprehension, reading, writing, and oral communication. An example of a physically active lesson related to these topics was ‘Mathematical Orienteering’, as previously described (for details see [16]). A similar game was ‘English Orienteering’, whereby physical and cognitive goals were the same, however, the English Language goals were to consolidate comprehension, communication of instruction, and direction using memorized expressions and phrases suitable for the situation. In this activity, each flag required the children to communicate complete sentences or reply to questions in English language. The answers to each problem provide children with a route around 18 stations, using 18 different exercises, with children writing the answers on a handout.

### 2.2. Participants

Twelve classes of a comprehensive Italian primary school were involved in this study. The classes were randomly assigned, six to the intervention group and six to the control group. Participants were 192 primary school children (mean age = 8.51; SD = 0.87 y), 94 in the intervention group (six classes; 39 males and 55 females) and 98 in the control group (six classes; 51 males and 47 females). Table 1 provides the socio-demographic variables of participants. Children in the experimental group received 8 additional hours of physical activity simultaneously with English Language and Mathematics activities for 16 weeks, whilst their peers in the control group followed normal curricular activities.

### 2.3. Measures

All measurements were made at two different time points: at the beginning (pre-intervention—January 2019) and end (post-intervention—June 2019) of the intervention. At each testing point, all tests were performed on the same day, and measurements were performed at the same time of day for each participant.

#### 2.3.1. Cognitive Function

The cognitive function tests were administered via the Weschler Intelligence Scale for Children, version IV (WISC-IV) [23]. The WISC-IV is a test battery for the neuropsychological evaluation comprising the main domains of cognitive function (language, visual perception, memory, attention, executive functions, reading, writing, and calculation) in children from 6 to 17 years old. Participants completed the tests on a one-to-one basis with a trained experimenter and have been used previously in a similar study population [16]. The testing battery took approximately 20 min to complete, and the tests were completed in the following order:

##### Digit Span

The Digit Span task comprised two tasks: digit span forwards (to assess attention and short-term memory) and digit span backwards (to assess verbal working memory) [24,25]. In both tasks, a series of digits (e.g., 3,8,6) were verbally presented to the participant, who was then asked to recall and verbally repeat the correct sequence. In the forward digit span participants repeat the digit in the same order as they were presented, whereas in the backward digit span participants were asked to repeat the digits in the reverse order. Participants were provided two opportunities at each sequence length. If the participant provided a correct answer for two sequences, the length of the list was increased by one digit. The test was terminated when participants failed two sequences at a particular sequence length. Each correctly recalled sequence was scored as one point, resulting in a score from 0 to 16 for each task (forwards and backwards).

##### Coding

The coding test measures visual-motor dexterity, associative nonverbal learning, and nonverbal short-term memory. Fine-motor dexterity, speed, accuracy, and ability to manipulate a pencil contribute to task success; perceptual organization is also important. Coding is a timed core Processing Speed subtest. For children aged 6–7 years old, the test is picture based. Children are given a worksheet with the first line containing the key. They must place a mark within all the other figures so that they match the key. The participants received one point for every correct answer, resulting in a score from 0 to 65. For children aged 8–16 years old, the key consists of boxes containing a numeral in the top line and a symbol in the bottom line. They must write the symbol corresponding to each numeral in the worksheet provided. The participants received one point for every correct answer, resulting in a score from 0 to 119.

##### Arithmetic Reasoning Ability

The Arithmetic test assessed the arithmetic reasoning ability of participants with a series of 34 problems of increasing difficulty. The participants had to mentally solve, without using paper and pencil or a calculator, each problem within 30 s. The test ended when the participant provided four consecutive wrong answers. The participants received one point for every correct answer, resulting in a score from 0 to 34. This arithmetic reasoning ability task has previously been used to assess the development of Mathematics learning in children [23]. 

#### 2.3.2. Gross Motor Skills

Gross motor skills were assessed with the Italian version of the Test of Gross Motor Development (third edition, TGMD-3) [26,27,28,29]. The TGMD-3 is divided into two sub-scales: locomotor and object control [29]. The locomotor skill sub-scale composed of six skills: run, gallop, hop, horizontal jump, slide (each judged on four performance criteria) and skip (judged on three criteria). The object control (ball skill) sub-scale was composed of seven skills: one hand forehand strike of self-bounced tennis ball, kick, overarm throw, underarm throw (each judged on four criteria), two-hand strike of a stationary ball (judged on five criteria), one hand stationary dribble of a basketball, and a two-handed catch (each judged on three criteria). A verbal description and demonstration of each skill was carried out by the examiner prior to the completion of the TGMD-3. Each participant completed three trials, one served as a practice, and then two main trials (for which data were recorded). Performances were observed and evaluated following the qualitative performance criteria for each TGMD-3 assessment skill, with each performance criteria scored as achieved (score awarded = 1) or not (score awarded = 0). The total score for each item was calculated by the sum of both trials for each skill, providing a locomotor subscale score, object control subscale score, and a total score (locomotor + object control).

### 2.4. Statistical Analysis

All data were analyzed using SPSS (SPSS v28.0; Chicago, IL, USA). Initially, baseline differences between the intervention and control groups were checked using an independent samples *t*-test. Subsequently, two-way mixed method (group (intervention vs. control) by time (time 1 and 2)) ANOVAs were performed for each outcome variable, with repeated measures for time. Cohen’s d effect sizes were calculated [30] and interpreted as per convention (negligible effect: ≥0 to <0.15; small effect: ≥0.15 to <0.40; medium effect: ≥0.40 to <0.75; large effect: ≥0.75 to <1.10; very large effect: ≥1.10 to <1.45; and huge effect: >1.45).

## 3. Results

### 3.1. Cognitive Function

Data for each of the cognitive function tests, pre- and post-intervention are displayed in Table 2. At baseline there were no significant differences between the intervention and control group for any cognitive variables (all *p* > 0.05). 

#### 3.1.1. Digit Span Forwards

Overall, there was a tendency for participants in the intervention group to score higher on the digit span forwards test compared to the control group (main effect of group, *F*(1,190) = 3.596, *p* = 0.059), and a tendency for improved performance over time (main effect of time, *F*(1,190) = 3.583, *p* = 0.060); though these did not reach statistical significance. However, the intervention group improved from pre- to post-intervention, whilst performance in the control group remained similar (group * time interaction, *F*(1,190) = 5.264, *p* = 0.023, *d* = 0.34 (small); Figure 2a).

#### 3.1.2. Digit Span Backwards

Overall, there was no difference in performance on the digit span backwards test between the intervention and control groups (main effect of group, *F*(1,190) = 2.117, *p* = 0.147), although performance improved over time (main effect of time, *F*(1,190) = 4.322, *p* = 0.039). Furthermore, the intervention group improved from pre- to post-intervention, whilst performance in the control group remained similar (group * time interaction, *F*(1,190) = 7.109, *p* = 0.008, *d* = 0.42 (medium); Figure 2b).

#### 3.1.3. Coding

Overall, participants in the intervention group scored higher on the coding test compared to the control group (main effect of group, *F*(1,190) = 7.263, *p* = 0.012), and performance improved over time (main effect of time, *F*(1,190) = 15.953, *p* < 0.001). Furthermore, the intervention group improved from pre- to post-intervention, whilst performance in the control group remained similar (group * time interaction, *F*(1,190) = 6.419, *p* = 0.012, *d* = 0.35 (small); Figure 2c).

#### 3.1.4. Arithmetic Reasoning

Overall, there was no difference in performance on the arithmetic reasoning test between the intervention and control groups (main effect of group, *F*(1,190) = 0.766, *p* = 0.383), although performance improved over time (main effect of time, *F*(1,190) = 11.120, *p* = 0.001). Furthermore, the improvement in performance in the intervention group from pre- to post-intervention was greater than in the control group (group * time interaction, *F*(1,190) = 5.333, *p* = 0.022, *d* = 0.40 (medium); Figure 2d).

### 3.2. Gross Motor Skills

Data for the TGMD-3 total score (and the locomotor and object control subscales) pre- and post-intervention are displayed in Table 3. At baseline there were no significant differences between the intervention and control group for total TGMD-3 score, or either subscale (all *p* > 0.05). 

#### 3.2.1. Total TGMD-3 Score

Overall, participants in the intervention group scored a higher total score for the TGMD-3 compared to the control group (main effect of group, *F*(1,190) = 45.437, *p* < 0.001), and performance improved over time (main effect of time, *F*(1,190) = 217.614, *p* < 0.001). Furthermore, the improvement in performance in the intervention group from pre- to post-intervention was greater than in the control group (group * time interaction, *F*(1,190) = 62.540, *p* < 0.001, *d* = 1.16 (very large); Figure 3a).

#### 3.2.2. Locomotor Subscale

Overall, participants in the intervention group scored a higher locomotor subscale score on the TGMD-3 compared to the control group (main effect of group, *F*(1,190) = 34.416, *p* < 0.001), and performance improved over time (main effect of time, *F*(1,190) = 86.084, *p* < 0.001). Furthermore, the improvement in performance in the intervention group from pre- to post-intervention was greater than in the control group (group * time interaction, *F*(1,190) = 11.299, *p* < 0.001, *d* = 0.63 (medium); Figure 3b).

#### 3.2.3. Object Control Subscale

Overall, participants in the intervention group scored a higher object control subscale score on the TGMD-3 compared to the control group (main effect of group, *F*(1,190) = 23.532, *p* < 0.001), and performance improved over time (main effect of time, *F*(1,190) = 149.513, *p* < 0.001). Furthermore, the improvement in performance in the intervention group from pre- to post-intervention was greater than in the control group (group * time interaction, *F*(1,190) = 69.821, *p* < 0.001, *d* = 1.29 (very large); Figure 3c).

## 4. Discussion

The main findings of the present study are that 16 weeks of physically active lessons, taught within the Mathematics and English Language curriculum, synergistically improved cognitive function and gross motor skill development in primary school children aged 8–9 years old. Improvements in cognitive function were seen across the domains of attention and memory, alongside improvements in arithmetic reasoning whilst improvements in gross motor skills were seen in both sub-elements (object control and locomotor skills) of the TGMD-3. 

The synergistic improvements in cognition and gross motor skill development in the present study are similar to those previously reported following a two-year implementation of physically active lessons [16]. However, the present study extends these findings by reporting similar beneficial effects in older children than the previous study (8–9 vs. 6–7 years old) and also over a much shorter time frame (16 weeks vs. 2 years). Furthermore, the present study involved the delivery of physically active lessons across both the Mathematics and English Language curriculum, compared to sole implementation within Mathematics in the previous study [16]. These findings are also in line with the meta-analytical evidence suggesting that short interventions (<8 weeks in duration) demonstrate significant benefits on educational outcomes [12]. Taken together, these findings have important implications for schools considering implementing physically active lessons, with the present study providing important evidence that: (i) co-delivery of physically active lessons across Mathematics and English Language teaching is effective; (ii) as little as 16-weeks delivery of physically active lessons enhances both cognition and gross motor skill development; and (iii) that these benefits exist in older children aged 8–9 years old, and thus physically active lessons are beneficial across the primary school age range.

When comparing the size of the effect with previous work, the effect sizes in the present study are smaller than those previously reported in previous work [16]. For example, in terms of cognitive outcomes, digit span forwards (*d* = 0.3 vs. 1.7), digit span backwards (*d* = 0.4 vs. 1.0), and arithmetic reasoning (*d* = 0.4 vs. 1.8) all demonstrated smaller effect sizes in the present study. Furthermore, in relation to gross motor skill development, total score (*d* = 1.2 vs. 1.7), locomotor skills (*d* = 0.6 vs. 1.1) and object control (*d* = 1.3 vs. 1.5) also demonstrated smaller effect sizes in the present study. We hypothesize that these differences are driven by a shorter duration intervention of physically active lessons in the present study. Interestingly, previous meta-analytical work has suggested overall effect sizes ranging from *d* = 0.3 [14,31] to *d* = 0.8 [12] for cognitive and educational outcomes. It is unsurprising, due to the shorter duration of intervention, that the effect sizes in the present study are smaller; however, the present study still demonstrates beneficial effects following a 16-week implementation. This could be a key message for schools considering implementation of physically active lessons, whereby benefits can be seen from as little as 16 weeks of implementation (which may be more realistic, at least initially in terms of real-world implementation). Future work could consider the effects of longer-term implementation of physically active lessons across both the Mathematics and English Language curriculum (perhaps throughout the duration of primary school), on cognition and gross motor skill development.

A key finding of the present study was the synergistic improvement in gross motor skill development, alongside the previously discussed cognitive benefits. The beneficial effects of physically active lessons on gross motor skill development are in line with previous work [16,32,33,34]. The improvements seen in gross motor skill development are of importance given that enhanced gross motor skills have been associated with greater well-being, fitness, and physical activity levels [22,34,35,36]. This is of particular importance given that, in Italy (the setting of the present study) as few as 7% of girls and 18% of boys aged 11 years old achieve physical activity recommendations [5]. Therefore, by enhancing gross motor skill learning through physically active lessons, it is possible that the benefits stretch beyond those achieved solely from the intervention itself, to ultimately enhance health and well-being in young people.

There are a number of plausible mechanisms that may underpin the synergistic benefits in cognition and gross motor skill development in the present study. For example, it has been suggested that participation in physical activity (as in the physically active lessons in the present study) leads to a more efficient allocation of neural resources [37,38]. Furthermore, specifically the combination of sensorimotor and cognitive processes, as required for physically active lessons, has been shown to enhance learning [39]. Whilst the present study did not specifically test these mechanisms, the observed effects add strength to the argument of their existence.

### 4.1. Limitations and Future Research Directions

The present study is not without limitation. For example, the duration of intervention in the present study was only 16 weeks, which is shorter than in some previous studies. In terms of cognition, the present study focused on attention, working memory, and arithmetic reasoning. Future work should look to explore the effects of physically active lessons across a wider range of domains of cognition, and for longer durations to fully understand the effects of the implementation of physically active lessons. Furthermore, in any applied study of this nature, it is not possible to control for all potential confounding variables which could influence cognition and academic achievement, such as the teaching provision and nutrition. Whilst every attempt was made to mitigate these variables, by including intervention and control participants within each school, the potential influence on study outcomes cannot be completely discounted. Finally, whilst previous work has reported beneficial effects of physically active lesson on academic achievement (e.g., [13]), this was not measured directly in the present study. Rather, it is postulate that the improvements seen in cognition will ultimately translate to enhanced learning and educational outcomes [40].

### 4.2. Conclusions

In conclusion, the present study demonstrates that 16 weeks of physically active Mathematics and Language lessons synergistically improved cognitive and gross moor skill outcomes in primary school children aged 8–9 years old. These findings demonstrate that implementation of physically active lessons over as little as 16 weeks can enhance the important outcomes of cognition, academic achievement and gross motor skill development. Therefore, based upon the findings of the present study, it should be recommended that school implement physically active lessons to enhance these important outcomes in their pupils; ultimately leading to enhanced educational outcomes and well-being.

## Figures and Tables

**Figure 1 ijerph-19-16751-f001:**
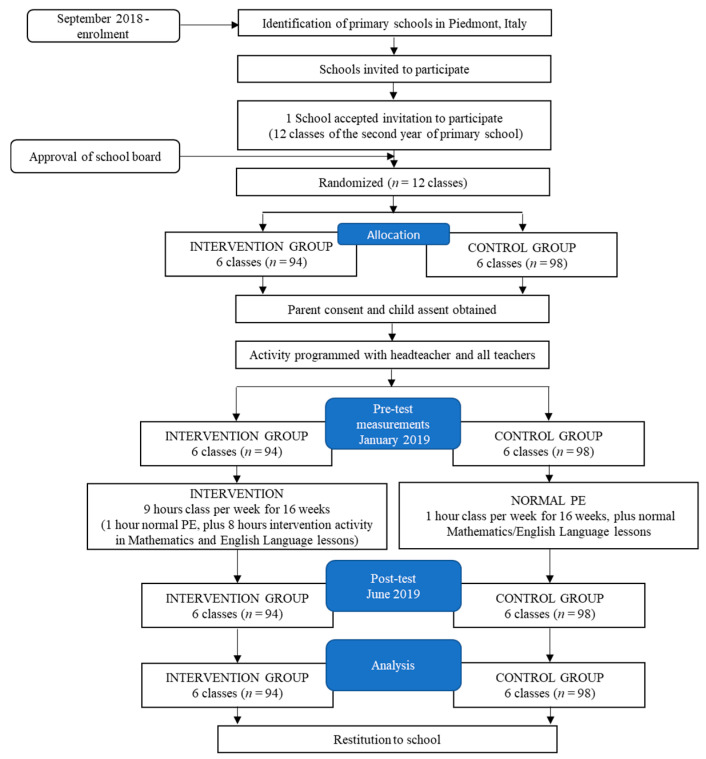
Experimental design and CONSORT flow diagram. (PE: Physical Education).

**Figure 2 ijerph-19-16751-f002:**
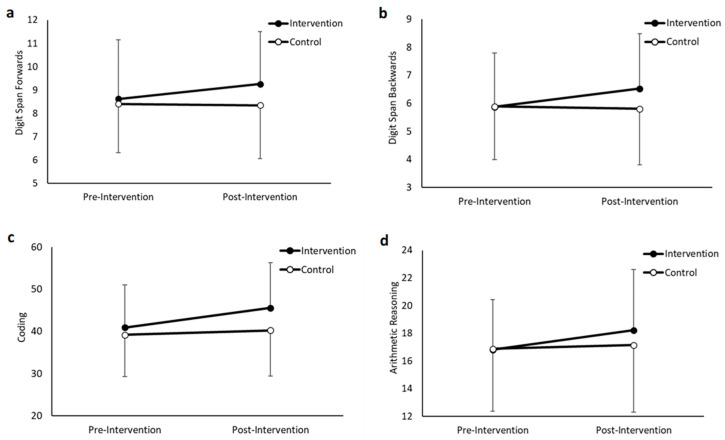
Performance on the cognitive function tests, pre- and post-intervention, in the intervention and control groups (data are mean ± SD). (**a**): Digit Span Forwards, group * time, *p* = 0.0.23; (**b**): Digit Span Backwards, group * time, *p* = 0.008; (**c**): Coding, group * time, *p* = 0.012; (**d**): Arithmetic Reasoning, group * time, *p* = 0.022.

**Figure 3 ijerph-19-16751-f003:**
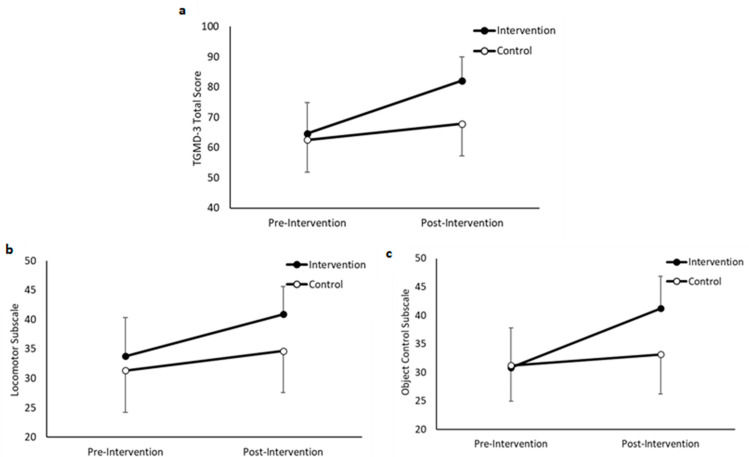
TGMD-3 total score (**a**), Locomotor subscale (**b**) and Object Control subscale (**c**), pre- and post- intervention, in the intervention and control groups (group * time interactions, all *p* < 0.001) (data are mean ± SD).

**Table 1 ijerph-19-16751-t001:** Descriptive characteristics of participants in the intervention and control groups. Data are mean ± SD.

	Intervention Group	Control Group	*p* Value ^a^
*n* (Males, Females)	94 (39, 55)	98 (51, 47)	-
Age [y]	8.5 ± 0.9	8.5 ± 0.8	0.864
Height [cm]	131.9 ± 7.7	132.6 ± 8.5	0.603
Body mass [kg]	31.6 ± 9.2	32.0 ± 8.8	0.819
Body mass index [kg m^−2^]	18.0 ± 4.4	17.9 ± 3.2	0.926

^a^ Independent samples *t*-test for comparison between groups.

**Table 2 ijerph-19-16751-t002:** Cognitive function data, pre- and post- intervention, in the intervention and control groups. Data are mean ± SD.

Cognitive Function Test	Intervention Group		Control Group		Group * Time Interaction	Cohen’s *d*
	Pre-Intervention	Post-Intervention	Pre-Intervention	Post-Intervention		
Digit Span Forwards	8.62 ± 2.54	9.26 ± 2.25	8.41 ± 2.10	8.35 ± 2.29	0.023	0.34
Digit Span Backwards	5.87 ± 1.94	6.53 ± 1.96	5.89 ± 1.89	5.81 ± 1.99	0.008	0.42
Coding	40.96 ± 10.05	45.56 ± 10.69	39.19 ± 9.90	40.25 ± 10.76	0.012	0.35
Arithmetic Reasoning	16.85 ± 3.62	18.22 ± 4.38	16.89 ± 4.50	17.14 ± 4.83	0.022	0.40

**Table 3 ijerph-19-16751-t003:** Gross motor skill data, pre- and post- intervention, in the intervention and control groups. Data are mean ± SD.

Test	Intervention Group		Control Group		Group * Time Interaction	Cohen’s *d*
	Pre-Intervention	Post-Intervention	Pre-Intervention	Post-Intervention		
TGMD-3 Total Score	65 ± 10	82 ± 8	63 ± 11	68 ± 11	<0.001	1.16
Locomotor Subscale	34 ± 7	41 ± 5	31 ± 7	35 ± 7	<0.001	0.63
Object Control Subscale	31 ± 7	41 ± 6	31 ± 6	33 ± 7	<0.001	1.29

## Data Availability

Data are available from the corresponding author upon reasonable request.

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
