# Peer review of "16 Weeks of Physically Active Mathematics and English Language Lessons Improves Cognitive Function and Gross Motor Skills in Children Aged 8–9 Years"

_ijerph, 2022, doi:10.3390/ijerph192416751_

Round 1
Reviewer 1 Report
This is a carefully executed study. While the results tend to confirm what we have always suspected - that physically active tasks in children improve learning and cognitive functioning - it is nice to see data that actually supports this. The text is well-written and clear throughout. I could raise no issues.
Author Response
Thank you for taking the time to review our paper and for the positive comments.
Reviewer 2 Report
This is an excellent and important paper. I think it would be improved by adding to the limitations section.
I believe that the limitations section is insufficient. It does not address other factors that contribute to cognitive outcomes including the teachers and the environment.
I think a discussion that included other factors that contribute to improved cognitive function including the quality of the teachers and environments should be considered.
Author Response
Thank you for taking the time to review our paper and for your positive comments. We agree that there are many other factors that could influence cognition within a study of this nature; such as the teaching within the schools and other variables such as nutrition. We have now included this in our amended limitations section [lines 448-453].
Reviewer 3 Report
The paper is well-written, clear and results are interesting and useful at the scientific and social levels.
Minor points
- Editing of the test is necessary (check for double spaces, typos, etc.)
- Write down what is “PE” in Figure 1 in the text or figure legend
- Since the “Institutional Review Board Statement” the manuscript looks not compiled by the authors and similar to the template. Please check these lines and correct.
- Please indicate what type of statistical errors the graphs show.
- Discussion could be ameliorated by increasing the references provided and discussing the results in term of the possible mechanisms that the intervention could have solicit.
Author Response
The paper is well-written, clear and results are interesting and useful at the scientific and social levels.
Thank you for taking the time to review our paper and for your positive comments.
Minor points
Editing of the test is necessary (check for double spaces, typos, etc.)
Apologies; the revised manuscript has been thoroughly checked prior to re-submission.
Write down what is “PE” in Figure 1 in the text or figure legend
Apologies for this oversight, this has been added to the Figure legend.
Since the “Institutional Review Board Statement” the manuscript looks not compiled by the authors and similar to the template. Please check these lines and correct.
Thank you; the ethical approval statement has been amended and moved to the start of the Methods section, along with the Journal specific requirements at the end of the manuscript.
Please indicate what type of statistical errors the graphs show.
Apologies for this oversight; the data are mean ± standard deviation. This information has been added to the figure legends.
Discussion could be ameliorated by increasing the references provided and discussing the results in term of the possible mechanisms that the intervention could have solicit.
Thank you for this useful suggestion. We have now added a short paragraph to the end of the Discussion to comment on some of the potential underlying mechanisms, alongside some additional supporting references.
Reviewer 4 Report
Authors presents 16 weeks of physically active mathematics and English language....
I have following major concerns.
i. Related work included but must be critical . The suggested effort should be framed in terms justification of this research.
ii. Explain dataset & methodology in further detail.
iii. The researchers should include reinforcement learning functionality in the workflow chart, as well as a short discussion of the complete figure and contributors.
iv. Describe main contributions in separate section after literature review section to show worth of your work.
v. How about the outcome of the data processing delay and its discussion?
vi. What objective does the planned work serve?
vii. Conclusion is too short.
Author Response
Authors presents 16 weeks of physically active mathematics and English language. I have following major concerns.
Related work included but must be critical . The suggested effort should be framed in terms justification of this research.
Thank you; we have highlighted in the amended introduction some critique of previous studies and how the present study makes a novel contribution to the literature by addressing some of these [lines 61-67, 73-76, 92-101, 104-108].
Explain dataset & methodology in further detail.
Thank you. We have reviewed the Methodology section and made a number of changes in response to your comments and those of the other reviewers and the Journal.
The researchers should include reinforcement learning functionality in the workflow chart, as well as a short discussion of the complete figure and contributors.
Thank you – the fact that physically active lessons were used to reinforce taught concepts has now been highlighted in the Methods section [lines 140-143]. Greater detail has also been added to figure legends in response to this comment and the comments of other reviewers. Finally, the author contribution statement has been completed.
Describe main contributions in separate section after literature review section to show worth of your work.
Thank you; in line with this comment and your comment above, we have highlighted in the amended introduction some critique of previous studies and how the present study makes a novel contribution to the literature by addressing some of these [lines 61-67, 73-76, 92-101, 104-108].
How about the outcome of the data processing delay and its discussion?
Thank you for highlighting this interesting point. We agree that there has been a delay between data collection and dissemination of this paper. However, unfortunately, not long after the completion of the study the COVID pandemic happened. This added significantly to academic workloads and has meant that we are only now in a position where we have found the time to be able to produce the paper. We do not feel that this delay affects the data, or our interpretation of the findings, in any way.
What objective does the planned work serve?
The aim of the present study is now highlighted in the amended manuscript [lines 97-101 ].
Conclusion is too short.
Thank you; some further detail has been added to the Conclusion section.
Round 2
Reviewer 4 Report
Revised version accepted